# MEA-Net: A Lightweight SAR Ship Detection Model for Imbalanced Datasets

**Yiyu Guo** 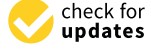 **and Luoyu Zhou ***

School of Electronics and Information, Yangtze University, Jingzhou 434023, China
* Correspondence: luoyuzh@yangtzeu.edu.cn

**Abstract:** The existing synthetic aperture radar (SAR) ship datasets have an imbalanced number of inshore and offshore ship targets, and the number of small, medium and large ship targets differs greatly. At the same time, the existing SAR ship detection models in the application have a huge structure and require high computing resources. To solve these problems, we propose a SAR ship detection model named mask efficient adaptive network (MEA-Net), which is lightweight and high-accuracy for imbalanced datasets. Specifically, we propose the following three innovative modules. Firstly, we propose a mask data balance augmentation (MDBA) method, which solves the imbalance of sample data between inshore and offshore ship targets by combining mathematical morphological processing and ship label data to greatly improve the ability of the model to detect inshore ship targets. Secondly, we propose an efficient attention mechanism (EAM), which effectively integrates channel features and spatial features through one-dimensional convolution and two-dimensional convolution, to improve the feature extraction ability of the model for SAR ship targets. Thirdly, we propose an adaptive receptive field block (ARFB), which can achieve more effective multi-scale detection by establishing the mapping relationship between the size of the convolution kernel and the channel of feature map, to improve the detection ability of the model for ship targets of different sizes. Finally, MEA-Net is deployed on the Jeston Nano edge computing device of the 2 GB version. We conducted experimental validation on the SSDD and HRSID datasets. Compared with the baseline, the AP of MEA-Net increased by 2.18% on the SSDD dataset and 3.64% on the HRSID dataset. The FLOPs and model parameters of MEA-Net were only 2.80 G and 0.96 M, respectively. In addition, the FPS reached 6.31 on the Jeston Nano, which has broad application prospects.

**Keywords:** synthetic aperture radar; ship detection; imbalanced datasets; data augmentation; attention mechanism; multi-scale detection

## 1. Introduction

Synthetic aperture radar (SAR) is a microwave imaging sensor capable of all-day, all-weather observations of the ground. Therefore, SAR is widely used in disaster investigation [1], environmental monitoring [2,3], target detection [4] and other fields. At present, the SAR ship detection has important research value in both the military field, which requires real-time position detection of specific ship targets, and the commercial field, which requires real-time scheduling of maritime transportation.

The traditional SAR ship detection mainly uses the constant false alarm rate algorithm (CFAR) [5–7] which is simple in calculation and has the characteristic of self-adaptation. The CFAR detector performs statistical modeling of the background clutter and adaptively estimates the threshold to determine whether the pixel is a ship or the background. It is suitable for scenes with high contrast between the ship and the background; however, the detection effect of ship targets in complex backgrounds is poor.

In recent years, with the development of the convolutional neural network (CNN), SAR ship detection models based on CNN have been widely studied due to their superior feature extraction capabilities. CNN-based object detection models are mainly divided into

two categories. One is one-stage, such as the "you only look once" (YOLO) series [8–10], single shot multibox detector (SSD) [11], CenterNet [12] and EfficientDet [13]; the other is two-stage, such as the region-CNN (R-CNN) series [14–17]. The target detection model based on two-stage is to generate a series of candidate areas by the algorithm, and then uses the CNN to classify these candidate regions. This model has high detection accuracy, but the model inference speed is slow. The one-stage target detection model abandons the candidate region generation stage, and directly performs target detection on the feature map extracted by the CNN, which can achieve faster model inference speed and better meet the needs of SAR ship detection.

Although the above methods have excellent performance, there are still some problems that need to be solved when they are directly applied to SAR ship detection. (1) The number of ship targets of different sizes in SAR images is imbalanced, and the features of small and large ship targets are very different. Therefore, a large number of missed and false detections occur in the SAR ship detection process, especially for small ship targets. (2) Unlike the targets in optical remote sensing images, ship targets in SAR images have weak texture, low contrast and speckle noise. Therefore, inshore ship detection is easily disturbed by the land, such as coastal ports and coasts. Moreover, in the common SAR ship dataset, the number of inshore ship targets is far lower than the number of offshore ship targets, which further reduces the detection accuracy of the model for inshore ship targets. (3) The structure of the SAR ship detection model is huge, consumes a lot of computing resources, and is difficult to deploy on the edge computing device.

Considering these existing problems, we propose a lightweight SAR ship detection model named MEA-Net for imbalanced datasets. It can effectively solve the problem of data imbalance, which includes an imbalanced number of inshore and offshore ship targets, and the number of small, medium and large ship targets. Finally, we deploy it on the Jeston Nano edge computing device of the 2 GB version.

The main contributions are as follows:

- Aiming at the problem that the number of inshore ship targets is far less than that of offshore ship targets, we propose a mask data balance augmentation (MDBA) method, with the purpose of solving the imbalance of inshore and offshore ship targets on the datasets.
- Aiming at the problem of low detection accuracy of inshore ship targets in SAR images, we propose a lightweight efficient attention mechanism (EAM) to enhance the ability of the model to extract the SAR ship target features. In this way, the important features of the ship target can be detected, and the influence of the complex background can be suppressed.
- Aiming at the problem of multi-scale ship target detection, we propose an adaptive receptive field block (ARFB) for multi-scale detection to improve the detection ability of the model for ship targets of different sizes. It can effectively fuse global features and local features by adapting the convolution kernel size of RFB-s module.

## 2. Related Works

Due to the increasing demand for SAR ship detection in the military and commercial fields, researchers in related fields have conducted extensive research. In this section, we will introduce the current research progresses in four aspects of SAR ship detection based on CNN, data augmentation method, attention mechanism and multi-scale detection.

### 2.1. SAR Ship Detection Based on CNN

Thanks to the excellent feature extraction capabilities of CNN, a series of target detection models based on CNN have emerged in recent years. Remote sensing target detection mainly includes optical-based [18] and SAR-based. Xu et al. [19] designed the multi-level alignment network including image-level, convolution-level and instance-level, to achieve cross-domain ship detection. Huang et al. [20] proposed a method based on transfer learning, which can learn knowledge from sufficient unlabeled SAR images to labeled

SAR images. Wang et al. [21] introduced the maximum stability extremal region (MSER) decision criterion to replace the original threshold criterion in Faster R-CNN target detection model. Wu et al. [22] proposed a new classifier and region-of-interest (RoI) feature integration strategy in Cascade R-CNN. Both of the above will further improve the accuracy of SAR ship detection. However, these two-stage target detection models have a large number of parameters and a slow inference speed, which is difficult to meet the needs of practical application deployment. For the lightweight deployment of the model, some researchers apply the one-stage target detection model to SAR ship detection, such as YOLO. Wang et al. [23] proposed a SAR ship detection model named SSS-YOLO which redesigned the feature extraction network based on YOLOv3 to improve the ability of the model to extract spatial and semantic information of small ship targets. Chen et al. [24] introduced the network pruning and knowledge distillation technology into YOLOv3 to realize the lightweight design of SAR ship detection model. Tang et al. [25] designed an N-YOLO ship detection model based on YOLOv5, using a noise level classifier which can effectively derive and classify the noise level of SAR images, to detect a SAR ship in a complex background with noise. In view of the above, this paper focuses on one-stage detectors, aiming to achieve more lightweight SAR ship detection.

### 2.2. Data Augmentation Method

Data augmentation in SAR images before passing them into model training can improve the robustness of the model. Jiang et al. [26] used the original image and two synthetic RGB polarimetric SAR images containing the contour information of ship as three color channels respectively, which reduced the effect of scattering noise and greatly enriched the training data. Sun et al. [27] proposed a random rotation Mosaic (RR-Mosaic) data augmentation method based on the Mosaic. The random angle rotation was introduced, which reduced the imbalance of angle categories. However, the existing research work on data augmentation methods does not focus on the imbalance between inshore and offshore ship targets. The MDBA method proposed in this paper solves this problem.

### 2.3. Attention Mechanism

The attention mechanism allows the model to learn the important features of the target while suppressing the secondary features, thereby improving the detection accuracy of the model. Cui et al. [28] proposed a spatial shuffle-group enhance (SSE) attention mechanism to solve the problem of confusion between ship targets and background clutter in large-scale SAR images. Zhu et al. [29] proposed a hierarchical attention mechanism. By introducing a global attention module (GAM) and a local attention module (LAM), the hierarchical attention strategy was proposed from the image layer and the target layer, respectively, thereby improving the important feature extraction ability of the model for ship targets in SAR images. Du et al. [30] designed a multi-scale feature attention module (MFAM), combining channel and spatial attention mechanisms, which can highlight important information and suppress interference caused by background clutter. Yang et al. [31] introduced a coordinate attention mechanism (CAM) to decompose channel attention into a two-dimensional feature encoding process, which aggregated features and captured long-range dependencies along two spatial directions, respectively. Ge et al. [32] designed a spatial orientation attention module (SOAM), which spliced a multi-receptive field spatial attention mechanism (MFSAM) on the CAM to make the model pay attention to the areas that need more attention, thereby enhancing the performance of ship detection in complex background. Cheng et al. [33] proposed the metric learning regularization which can enforce the model to be more discriminative to better focus on the features of different classes. The attention mechanisms proposed above all can achieve good performance. However, after being embedded into the model, they have a great impact on model complexity. Therefore, the EAM proposed in this paper takes into account both the accuracy and the lightweight design, reducing the impact on the model complexity.

### 2.4. Multi-Scale Detection

On the SAR ship dataset, the proportion of ship targets of different sizes is imbalanced. Therefore, some researchers have studied multi-scale detection methods to improve the detection ability of ship targets of different sizes. Multi-scale detection mainly realizes two parts of optimization, namely multi-scale feature fusion and the receptive field increasement. Liu et al. [34] embedded the receptive field block (RFB) into YOLOv4 to solve the problem of SAR ship scale diversity, which improved the performance of the model to detect multi-scale ship targets. Yu et al. [35] designed a FASC-Net which increased the receptive field of the feature map through the spatial pyramid pooling (SPP) module and performed feature fusion through the channel-attention path enhancement (CAPE) module to better predict SAR ship targets of different sizes. Wei et al. [36] proposed a high-resolution feature pyramid Network (HRFPN) which utilized feature maps of high-resolution and low-resolution convolutions to achieve more accurate multi-scale detection. Zhang et al. [37] proposed a quad feature pyramid network (quad-FPN) to gradually improve the SAR ship detection performance by connecting four types of FPN networks. The multi-scale detection mechanisms proposed above all have achieved good performance, but there is still room for further optimization. Based on the principle of receptive field in brain neurons, this paper proposes an adaptive receptive field method to further improve the accuracy of multi-scale detection.

## 3. Materials and Methods

### 3.1. Overall Framework

MEA-Net is designed on the baseline YOLOX-Nano [38] which is lightweight and has strong generalization ability. The overall framework is shown in Figure 1.

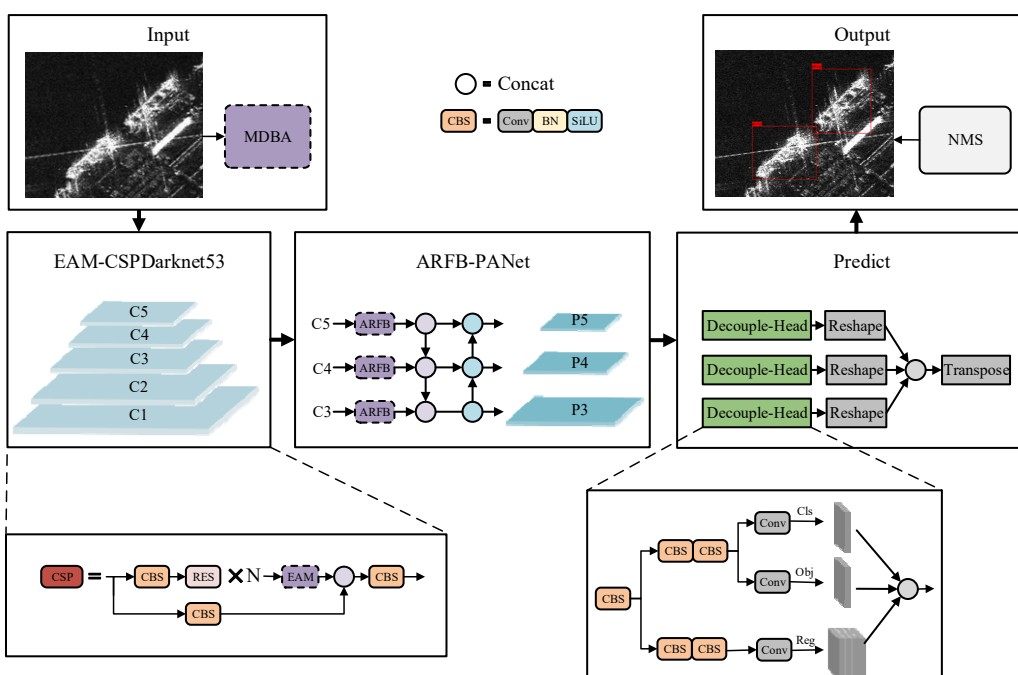

**Figure 1.** The overall framework of MEA-Net, where the purple dashed boxes represent improved parts.

Firstly, in the input part, we propose a mask data balance augmentation (MDBA) method which can effectively solve the data imbalance problem of inshore and offshore ship targets on the dataset to enhance the ability of the model to detect ship targets in different positions and backgrounds. Secondly, to improve the feature extraction capability of the model, we propose a lightweight efficient attention mechanism (EAM) module to be

embedded into the CSPDarknet53 backbone network. Thirdly, at the front of the PANet neck network, we propose an adaptive receptive field block (ARFB) for multi-scale detection which fuses local and global features during feature extraction to enhance the ability of the model to detect objects of different sizes. Finally, the Decouple-Head technology is used to achieve accurate detection, and the non-maximum suppression (NMS) algorithm is used to eliminate redundant detection boxes to find the correct position of the SAR ship targets.

### 3.2. MDBA Method

SAR inshore ship targets are easily affected by land, such as ports and coasts, which results in low accuracy and high missed detection rates for inshore ship targets. Therefore, we propose a MDBA method to solve the large gap in the amount of inshore and offshore SAR ship targets on the dataset. In the process of data augmentation, if there are both inshore and offshore ship targets in the SAR image, mask windows are used to remove the offshore targets and then transfer into the model for training. In this way, the number of inshore ship targets during model training is increased, while the number of offshore ship targets is suppressed. The algorithm steps are shown in Figure 2. We pass the SAR image into the MDBA method to determine whether it contains land. If land is not included, data augmentation is not performed in this SAR image. If land is included, after traversing all the ship targets in the entire image, the inshore ship targets are kept and the offshore ship targets are removed through mask windows. After that, the image is flipped and rotated at different angles for data augmentation. There are two important links here, one is to determine whether the SAR image contains land, and the other is to determine whether the ship target is inshore or offshore.

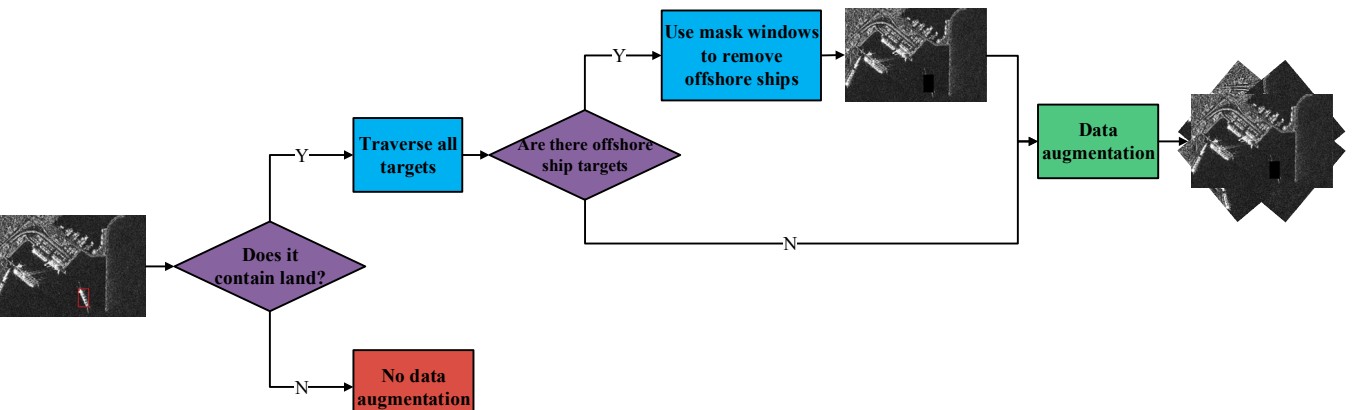

**Figure 2.** MDBA method flow chart.

The algorithm steps for judging whether the SAR image contains land are shown in Figure 3. Firstly, the original SAR image and extensible makeup language (XML) label files are obtained. And the original image is binarized to obtain a binary image $f(x,y)$. Secondly, we assign zero to the target area in the binarized image according to the XML label to obtain a new SAR image $f'(x,y)$ without the ship target. After the image is subjected to morphological processing of dilate and erosion, the flood fill algorithm (*FFA*) is used to fill in the holes of the land area to obtain a filtered image $h(x,y)$. This process can be described as,

$$h(x,y) = FFA\big[(f'(x,y) \odot B) \Theta B\big] \tag{1}$$

where $\odot$ represents the dilate operation, $\Theta$ represents the erosion operation, $B$ represents the image template used in the dilate and erosion process, and $FFA[\cdot]$ represents the flood filling algorithm operation. We set $C_p$ to represent whether the $p$-th SAR image contains land, and specify a threshold $T_m$. Here, $T_m$ is associated with the image resolution. Then,

the number of pixels of the maximum connected region is calculated. If the value is greater than $T_m$, it is judged that land is in the SAR image. This process can be described as,

$$C_p = \begin{cases} 1 & L[h(x,y)] > T_m \\ 0 & else \end{cases} \tag{2}$$

where $L[\cdot]$ represents the number of pixels for obtaining the maximum connected region.

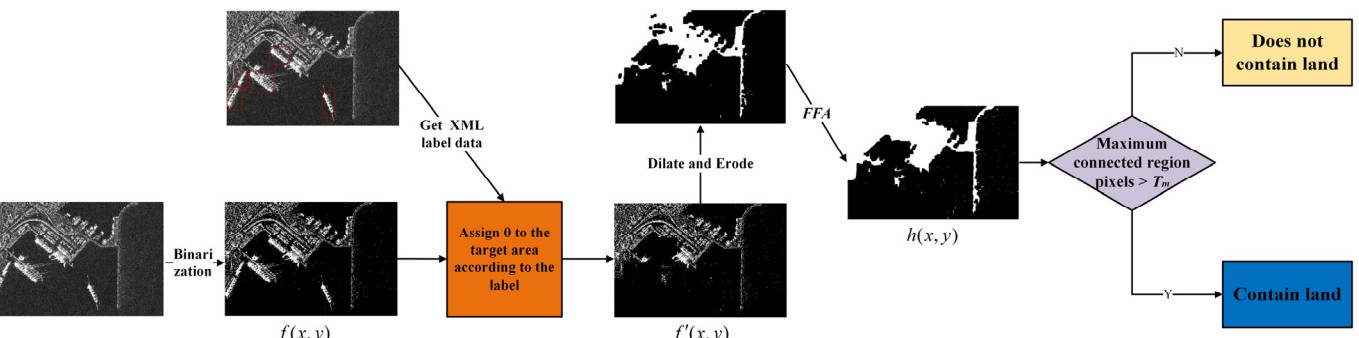

**Figure 3.** Steps to determine whether the SAR image contains land.

The steps of judging whether the ship target is inshore or offshore are shown in Figure 4. Firstly, we get the final image $h(x,y)$ from the previous step and the mask image $g(x,y)$ which is generated from the original image and the corresponding XML label files. Secondly, the center point of each rectangular block (i.e., the ship target) in the mask image is calculated to generate the set $g'(x,y)$. Thirdly, the threshold $T_m$ of the previous step is used to remove the land area smaller than it, and then the edge information is extracted by the Canny operator [39] to generate the land boundary points set $h'(x,y)$. Finally, the distance between each element in $g'(x,y)$ and $h'(x,y)$ is calculated to form a two-dimensional array $D$, and the calculation of each of its elements is shown in Formula (3).

$$D_{i,j} = \sqrt{\left(x_i - x_j\right)^2 + \left(y_i - y_j\right)^2} \ 1 \leq i \leq N, 1 \leq j \leq M \tag{3}$$

where $N$ is the length of set $g'(x,y)$ and $M$ is the length of set $h'(x,y)$.

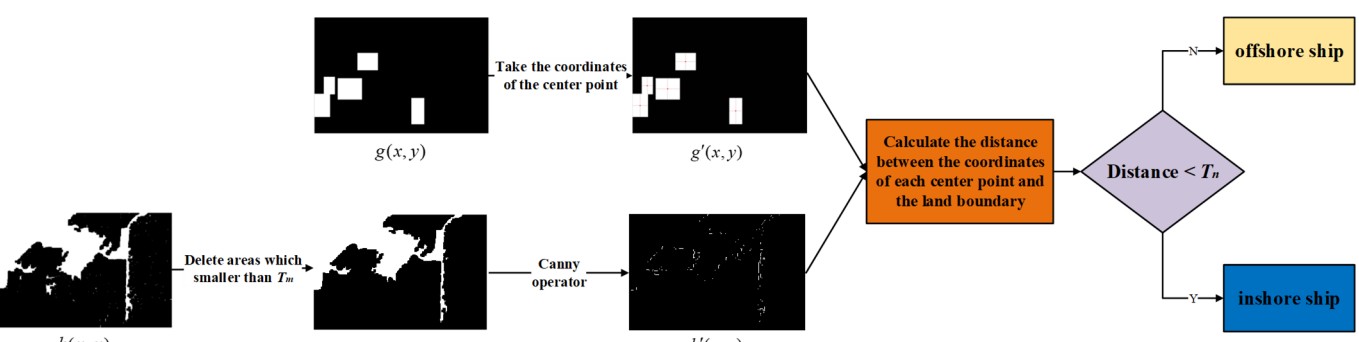

**Figure 4.** Steps to determine whether the ship target is inshore or offshore.

We set $G_q$ to represent whether the $q$-th ship is the inshore ship and specify a threshold $T_n$ which is related to the ship target size. Here, we define $T_n = kP_q$, where $P_q$ is the number of pixels for the $q$-th ship and $k$ is the scale factor. Then we calculate the $P_q$ to get the corresponding $T_n$ and take the minimum element of the $q$-th row of $D$. If it is less than the threshold $T_n$, the target is judged to be an inshore ship target. This process is described as,

$$G_q = \begin{cases} 1 & Min[D_{q,*}] < T_n \\ 0 & else \end{cases} \tag{4}$$

where $Min[\cdot]$ represents the operation of obtaining the minimum element of the row matrix.

### 3.3. EAM Module

The attention mechanism can make the model pay more attention to the important features of the ship target. It highlights the effective information and suppresses the invalid information. However, the introduction of attention mechanism inevitably brings additional computational burden. Inspired by the attention mechanisms of CBAM [40] and ECA [41], we propose a low-computation and high-accuracy hybrid domain attention mechanism EAM for extracting important features of SAR ship targets. Its structure diagram is shown in Figure 5.

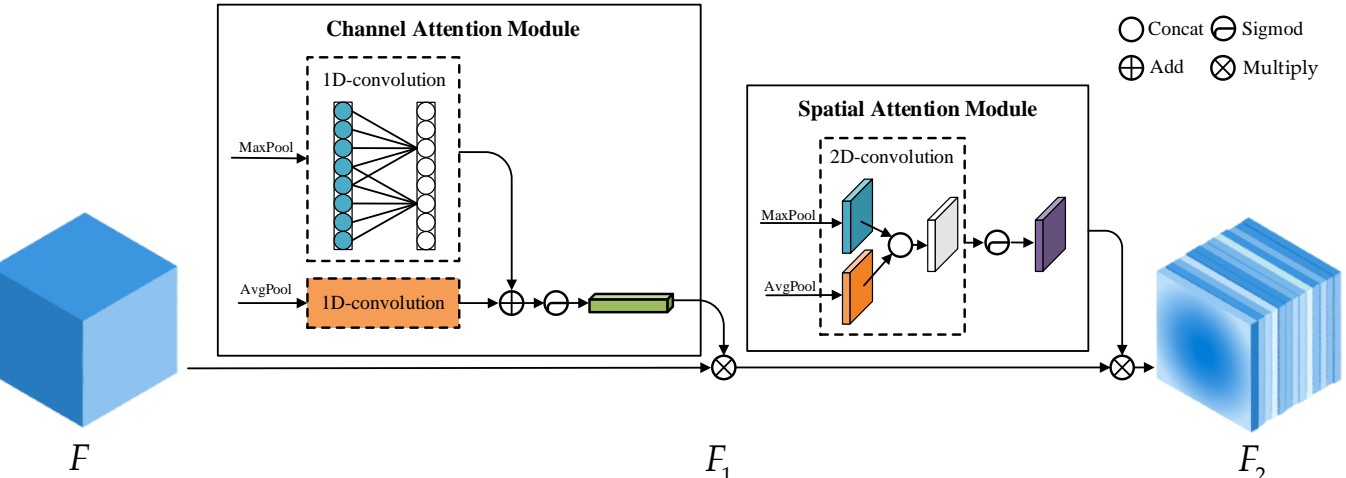

**Figure 5.** EAM module structure diagram.

The EAM module is composed of the channel attention mechanism and the spatial attention mechanism in series, which can be defined as:

$$\begin{cases} F_1 = M_{CAM}(F) \otimes F \\ F_2 = M_{SAM}(F_1) \otimes F_1 \end{cases} \tag{5}$$

where $M_{\text{CAM}}(\cdot)$ represents channel attention operation, $M_{\text{SAM}}(\cdot)$ represents spatial attention operation, and $\otimes$ represents weighted multiplication operation of characteristic graph.

The channel attention part of the EAM module uses one-dimensional convolution to obtain the channel information of the feature map. This method has the following three characteristics: (1) The cross-channel interaction is realized. (2) The one-to-one correspondence between channel attention weights and channels is preserved. (3) The amount of computation and parameters is low, thereby reducing the complexity of the model after embedding the attention mechanism module. The implementation of the channel attention part is as follows. Firstly, the input feature map is subjected to global maximum pooling and global average pooling. Secondly, the one-dimensional convolution operation with a convolution kernel length of $k$ is performed (here, $k$ is associated with the number of channels, $C$), and the elements at the corresponding positions are added one by one. Thirdly, the Sigmod activation function normalizes the values between zero and one. Finally, the feature map $F_1$ is obtained by the weighted multiplication with the result of the previous step and the input feature map, and $F_1$ is passed to the spatial attention module. The channel attention process can be described as the following formula,

$$M_{CAM}(F) = \sigma\left(C1D_k\left(F_{avg}^c\right) + C1D_k(F_{\max}^c)\right) \tag{6}$$

$$k = \psi(C) = \left| \frac{\log_2(C)}{2} + \frac{1}{2} \right|_{odd} \tag{7}$$

where $C1D_k(\cdot)$ represents the one-dimensional convolution operation of convolution kernel size $k$, $F_{avg}^c$ is the channel characteristic diagram obtained by the global average pooling of $F$, $F_{max}^c$ is the channel characteristic diagram obtained by the global maximum pooling of $F$, $\sigma(\cdot)$ represents Sigmod activation function operation, and $|\cdot|_{odd}$ represents the closest odd number operation.

The spatial attention part of the EAM module uses two-dimensional convolution to obtain the spatial information of the feature map. Firstly, the global maximum and average pooling are performed on the intermediate feature map $F_1$, and then are concatenated. Secondly, the two-dimensional convolution is performed to obtain the feature map of $1 \times H \times W$. Thirdly, the Sigmod activation function normalizes the values between zero and one. Finally, the feature map $F_2$ is obtained by the weighted multiplication with the result of the previous step and the intermediate feature map $F_1$. The spatial attention process can be described as,

$$M_{SAM}(F_1) = \sigma\left(C2D_7\left(F_{1avg}^S; F_{1max}^S\right)\right) \tag{8}$$

where $C2D_7(\cdot)$ represents the two-dimensional convolution operation of the convolution kernel $7 \times 7$, $F_{1avg}^S$ is the spatial characteristic diagram obtained by the global average pooling of $F_1$, and $F_{1max}^S$ is the spatial characteristic diagram obtained by the global maximum pooling of $F_1$.

### 3.4. ARFB Module

In order to improve the detection ability of the model for SAR ship targets of different sizes, we propose an adaptive multi-scale detection module ARFB. In the visual cortex, the size of the receptive field of neurons in the same area is different, which makes the neurons collect multi-scale spatial information in the same stage. The RFB-s module [42] is built for this reason.

The RFB-s module is shown in Figure 6a, which has four branches and shortcut connections. Each branch first reduces the dimension through $1 \times 1$ convolution to reduce the amount of computation, and then obtains receptive fields of different sizes through convolution, asymmetric convolution, and dilation convolution. Here, dilated convolutions with three different dilation rates of 1, 3, and 5 are used. The four branches are concatenated in the channel dimension, pass through $1 \times 1$ convolution, and merge with the shortcut. Finally, the output is obtained through the Relu activation function.

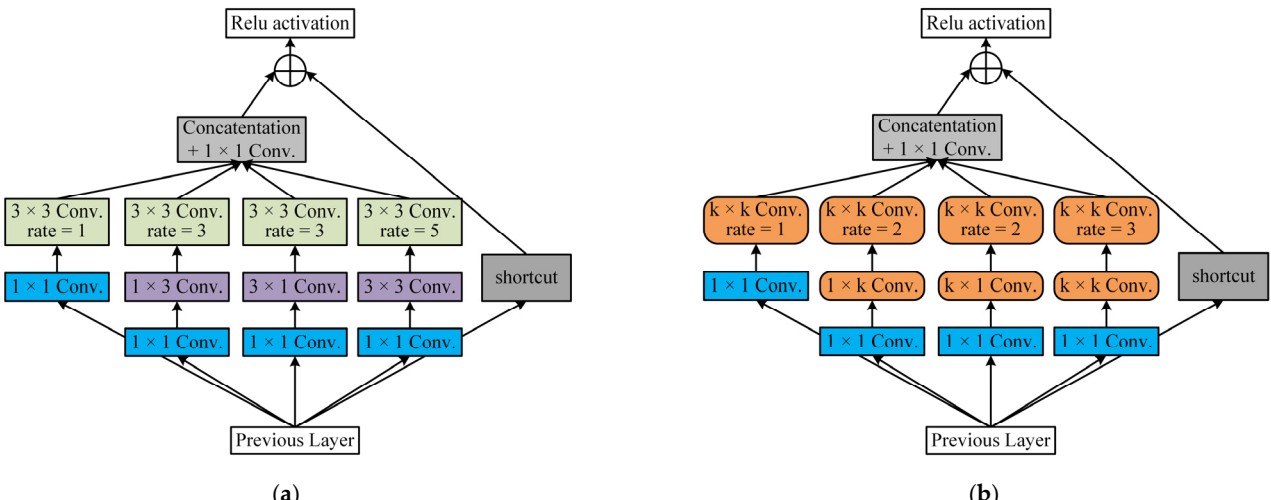

**Figure 6.** Multi-scale detection module. (**a**) RFB-s. (**b**) ARFB.

The RFB-s, which is based on the inception structure, obtains feature maps by stacking convolution kernels of different sizes. SKNet [43] points out that this is insufficient, and

the model should dynamically adjust the size of the convolution kernel to adjust its own receptive field. Based on this idea, we propose an adaptive multi-scale detection module named ARFB. Its structure is shown in Figure 6b.

Based on the RFB-s module, we establish a mapping relationship between the convolution kernel parameter $k$ and the multiplication of the height and width (i.e., $H \times W$) of the feature map. The mapping relationship here needs to meet two conditions. One is monotonically increasing, and the other is non-linear. Therefore, we introduce a unary quadratic function to represent the mapping relationship, as shown in Formula (9),

$$H \times W = \varphi(k) = ak^2 - b(k > 0) \tag{9}$$

Given the height ($H$) and width ($W$) of the feature map, the value of $k$ can be uniquely determined, as shown in Formula (10),

$$k = \psi(H \times W) = \left| \sqrt{\frac{H \times W + b}{a}} \right|_{\text{odd}} \tag{10}$$

where $|\cdot|_{\text{odd}}$ represents the closest odd number operation. The selection of $a$ and $b$ needs to consider two aspects. One is that it is not suitable to use too large a convolution kernel in the feature map of small resolution, and the other is that the size of the convolution kernel should be relatively smooth with the change of the $H \times W$ of the feature map. Therefore, in this paper, we set $a$ and $b$ to 400 and 5000, respectively.

The ARFB module is added after the three output feature maps C3, C4, and C5 of the backbone network. After the calculation, the $k$ value of the convolution kernel of the ARFB module introduced into each feature map is shown in Table 1.

**Table 1.** The convolution kernel $k$ value of ARFB module introduced in different feature maps.

| Feature Map | Input Size | $k$ |
|:---:|:---:|:---:|
| C3 | $80 \times 80$ | 5 |
| C4 | $40 \times 40$ | 5 |
| C5 | $20 \times 20$ | 3 |

## 4. Results

### 4.1. Experimental Datasets and Platforms

The performance test of the model in this paper is implemented using the SSDD dataset, and the migration test of the model is implemented using the HRSID dataset. The data distribution of the two datasets is shown in Figure 7.

The SSDD dataset [44] is derived from RadarSat-2, TerraSAR and Sentinel-1, with a resolution between 1 m and 15 m, including 1160 SAR images. There are 2456 ship targets, of which small ship targets account for 60.2%, medium ship targets account for 36.8%, and large ship targets account for 3%. The HRSID dataset [45] is derived from Sentinel-1 and Terra-SAR-X, with a resolution of 0.5 m, 1.0 m and 3.0 m, including 5604 SAR images. There are 16,591 ship targets, of which small ship targets account for 54.5%, medium ship targets account for 43.5%, and large ship targets account for 2%.

In this paper, we train the model on the Windows platform and evaluate the metrics on the Jeston Nano edge computing device. The GPU of the Windows platform is NVIDIA GeForce RTX 3070, the CPU is AMD Ryzen7 3700X, and the memory is 16 G. In the training process, the input resolution is $640 \times 640$, the initial learning rate is 0.001, and the learning rate is updated by the Adam algorithm. The GPU of Jetson Nano is Maxwell with 128 cores, the CPU is ARM A57 with 4 cores and a frequency of 1.43 G, the memory is 2 GB, and the power supply is 10 W.

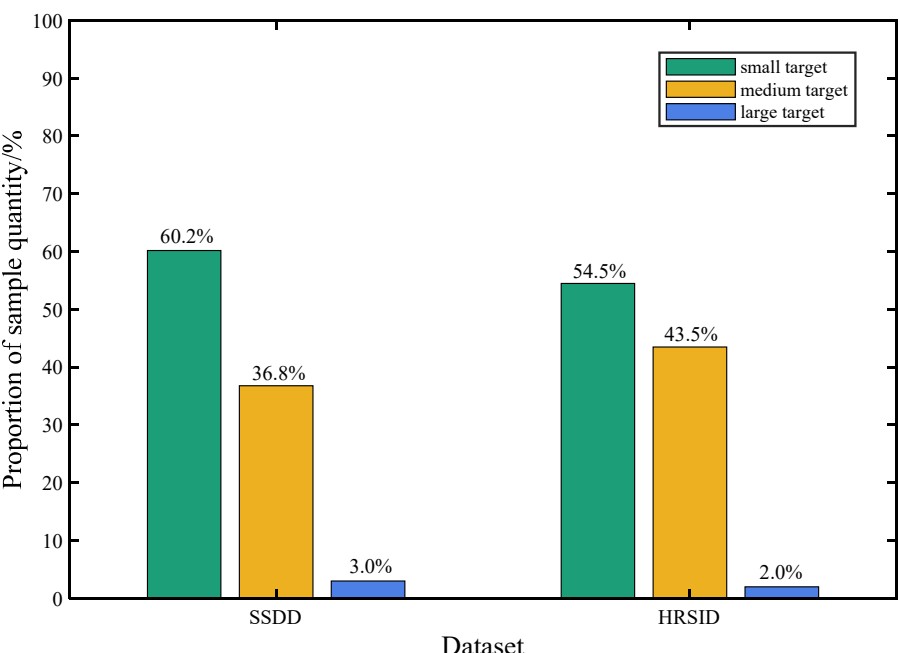

**Figure 7.** Data distribution of the SSDD and HRSID datasets.

## 4.2. Evaluation Metrics

For the detection of SAR ship targets, it is necessary to comprehensively consider the detection accuracy, model complexity and real-time performance. Therefore, the experiments in this paper use the precision rate ($P$), recall rate ($R$), average precision ($AP$), $AP$ of small targets ($AP_S$), $AP$ of medium targets ($AP_M$), $AP$ of large targets ($AP_L$), and $F1$ as the evaluation indicators of the model detection accuracy. Defined by the following formula,

$$P = \frac{TP}{TP + FP} \tag{11}$$

$$R = \frac{TP}{TP + FN} \tag{12}$$

$$AP = \int_0^1 P(R)dR \tag{13}$$

$$F1 = \frac{2 \times P \times R}{P + R} \tag{14}$$

where $TP$ represents the number of true positives, $FP$ represents the number of false positives, $TN$ represents the number of false negatives, and $P$ ($R$) represents the ordinate is the precision rate and the abscissa is the curve made by the recall rate, respectively. In this paper, $AP$, $AP_S$, $AP_M$ and $AP_L$ of all experiments are calculated under the condition of intersection over union (IoU) = 0.5.

Model complexity is evaluated using the floating-point operations (FLOPs) and model parameters. The real-time performance is evaluated using frame per second ($FPS$), which is as follows:

$$FPS = \frac{N}{T} \tag{15}$$

where $N$ is the number of SAR images in the test set, and $T$ is the seconds required for detection when the model is deployed on the Jeston Nano edge computing device of the 2 GB version.

*4.3. The Effect of the MDBA Method*

In order to verify the effectiveness of the MDBA method proposed in this paper, experiments are performed on the baseline, the model introducing the data augmentation (DA) which includes flip and rotation at different angles, and the model introducing the MDBA method. SSDD and HRSID datasets are used to test, and the results are shown in Table 2. Here, we divide the test set into two parts: inshore ship test set and offshore ship test set. On the SSDD dataset, after introducing the MDBA method, the $AP$ and $AP_i$ of the model increases, respectively, by 1.17% and 7.58%, while the $AP_o$ of the model only decreases by 0.17%. However, after introducing the DA method, the $AP$, $AP_i$ and $AP_o$ of the model only increase, respectively, by 0.30%, 1.36% and 0.28%. Therefore, the detection accuracy improvement of introducing DA is much lower than that of introducing MDBA, especially for inshore ship targets. On the HRSID dataset, the detection accuracy of the introduction of MDBA is also much better than the baseline and the introduction of DA, especially for inshore ship targets. The $AP_i$ of the introduction of MDBA is 7.19% and 4.35% higher than the baseline and the introduction of DA, respectively. Thus, the effectiveness of the MDBA data augmentation method is demonstrated.

**Table 2.** The effect of the MDBA method.

| Dataset | Model | *P* (%) | *R* (%) | *F1* (%) | *AP* (%) | $AP_i$ [1] (%) | $AP_o$ [2] (%) |
|---------|-------|---------|---------|----------|----------|----------------|----------------|
| SSDD | baseline | 92.04 | 87.85 | 89.90 | 90.65 | 67.60 | 95.52 |
| | +DA | 93.99 | 87.09 | 90.41 | 90.95 | 68.96 | 95.80 |
| | +MDBA | 93.52 | 89.49 | 91.46 | 91.82 | 75.18 | 95.35 |
| HRSID | baseline | 94.26 | 73.72 | 82.73 | 85.42 | 66.56 | 93.26 |
| | +DA | 93.66 | 74.62 | 83.07 | 85.55 | 69.40 | 92.80 |
| | +MDBA | 94.13 | 76.93 | 84.67 | 87.51 | 73.75 | 93.76 |

[1] $AP_i$ represents the $AP$ of inshore ship test set. [2] $AP_o$ represents the $AP$ of offshore ship test set.

*4.4. The Effect of the EAM Module*

In order to verify the effectiveness of the EAM attention mechanism proposed in this paper, it is compared with commonly used attention mechanisms such as CBAM [40], ECA [41], CA [46] and SE [47]. The results are shown in Table 3. Firstly, after the introduction of the different attention mechanisms, the accuracy of the model improves. After the introduction of the EAM attention mechanism, the $F1$ and $AP$ improve the most, increasing by 1.00% and 1.09% respectively. Secondly, the introduction of the EAM only needs to add 812 parameters to the model, which is far less than the 5920, 13,920, and 6704 parameters added by the introduction of SE, CA and CBAM attention mechanisms. Although the introduction of the ECA attention mechanism only needs to increase the 28 parameters of model, the $F1$ and $AP$ are 1.14% and 0.63% lower than the introduction of EAM, respectively. Parameter changes of this order of magnitude have little effect on the inference speed of the model, while 1.14% $F1$ and 0.63% $AP$ reduction have a certain degree of impact on model accuracy. In summary, after embedding the EAM attention mechanism, the model achieves the most balanced effect in $F1$, $AP$, and added parameters.

**Table 3.** Comparison of the effects of different attention mechanisms.

| Model | Added Parameters | *P* (%) | *R* (%) | *F1* (%) | *AP* (%) |
|-------|------------------|---------|---------|----------|----------|
| baseline | - | 92.04 | 87.85 | 89.90 | 90.65 |
| +SE | 5920 | 93.03 | 86.20 | 89.49 | 91.11 |
| +ECA | 28 | 92.60 | 87.09 | 89.76 | 91.18 |
| +CA | 13,920 | 94.58 | 86.20 | 90.20 | 91.37 |
| +CBAM | 6704 | 94.43 | 85.82 | 89.92 | 91.05 |
| +EAM | 812 | 94.90 | 87.22 | 90.90 | 91.74 |

### 4.5. The Effect of the ARFB Module

In order to verify the effectiveness of the ARFB multi-scale detection module proposed in this paper, the baseline; the RFB-s module added after the C3, C4 and C5 feature maps; and the ARFB module added after the C3, C4, and C5 feature maps are compared. The results are shown in Table 4. It can be seen that after the introduction of the ARFB module, the *F1*, *AP*, *AP_S*, *AP_M* and *AP_L* improve by 0.84%, 1.46%, 0.80%, 2.12% and 2.87%, respectively, compared with the baseline. After the introduction of ARFB, the detection ability of the model for small, medium and large ship targets improves to varying degrees. After the introduction of the RFB-s module, although the *AP* of the model increases by 0.96%, the detection accuracy of small targets decreases by 2.01%. This proves that the adaptive convolution kernel size method of ARFB module is effective in improving the detection accuracy of model for ship targets of different sizes.

**Table 4.** Comparison of the effects of different multi-scale detection module.

| Model | *P* (%) | *R* (%) | *F1* (%) | *AP* (%) | $AP_S$ (%) | $AP_M$ (%) | $AP_L$ (%) |
|---|---|---|---|---|---|---|---|
| baseline | 92.04 | 87.85 | 89.90 | 90.65 | 88.86 | 93.15 | 89.09 |
| +RFB-s | 93.70 | 86.58 | 90.00 | 91.61 | 86.85 | 95.34 | 89.11 |
| +ARFB | 94.27 | 87.47 | 90.74 | 92.11 | 89.66 | 95.27 | 91.96 |

$AP_S$, $AP_L$ and $AP_M$ are calculated at IoU = 0.5.

### 4.6. Ablation Experiments

In this section, we design a series of experiments to verify the effects of the MDBA method, the EAM module and the ARFB module on the model performance, respectively, and the results are shown in Table 5. It can be seen that the introduction of the MDBA method can improve the *AP* of the model by 1.17% without increasing the amount of computation. The introduction of EAM and ARFB only needs to increase the calculation amount of 0.33 G FLOPs and 0.01 G FLOPs, which can increase the *AP* by 1.09% and 1.46%, respectively. After embedding MDBA, EAM and ARFB at the same time, the model achieves the best results. The *P*, *R*, *F1* and *AP* improve by 3.51%, 2.58%, 3.02% and 2.18% compared with the baseline. It can be proved that the designed modules in this paper are all effective, feasible and will not have too much influence on the complexity of the model.

**Table 5.** The results of ablation experiments.

| MDBA | EAM | ARFB | *P* (%) | *R* (%) | *F1* (%) | *AP* (%) | FLOPs (G) |
|---|---|---|---|---|---|---|---|
| × | × | × | 92.04 | 87.85 | 89.90 | 90.65 | 2.46 |
| √ | × | × | 93.52 | 89.49 | 91.46 | 91.82 | 2.46 |
| × | √ | × | 94.90 | 87.22 | 90.90 | 91.74 | 2.79 |
| × | × | √ | 94.27 | 87.47 | 90.74 | 92.11 | 2.47 |
| √ | √ | √ | 95.55 | 90.43 | 92.92 | 92.83 | 2.80 |

### 4.7. Visualization Experiment

In this section, we demonstrate the performance of MEA-Net by means of predicted boxes and Grad-CAM [48] heatmaps. Grad-CAM is able to visualize the feature extraction effect of the model. This technology can provide a better visual explanation for the performance of the neural network-based model and give a view of interpretability.

Figure 8 shows the detection effect of the baseline model and the MEA-Net model proposed in this paper on the SSDD dataset.

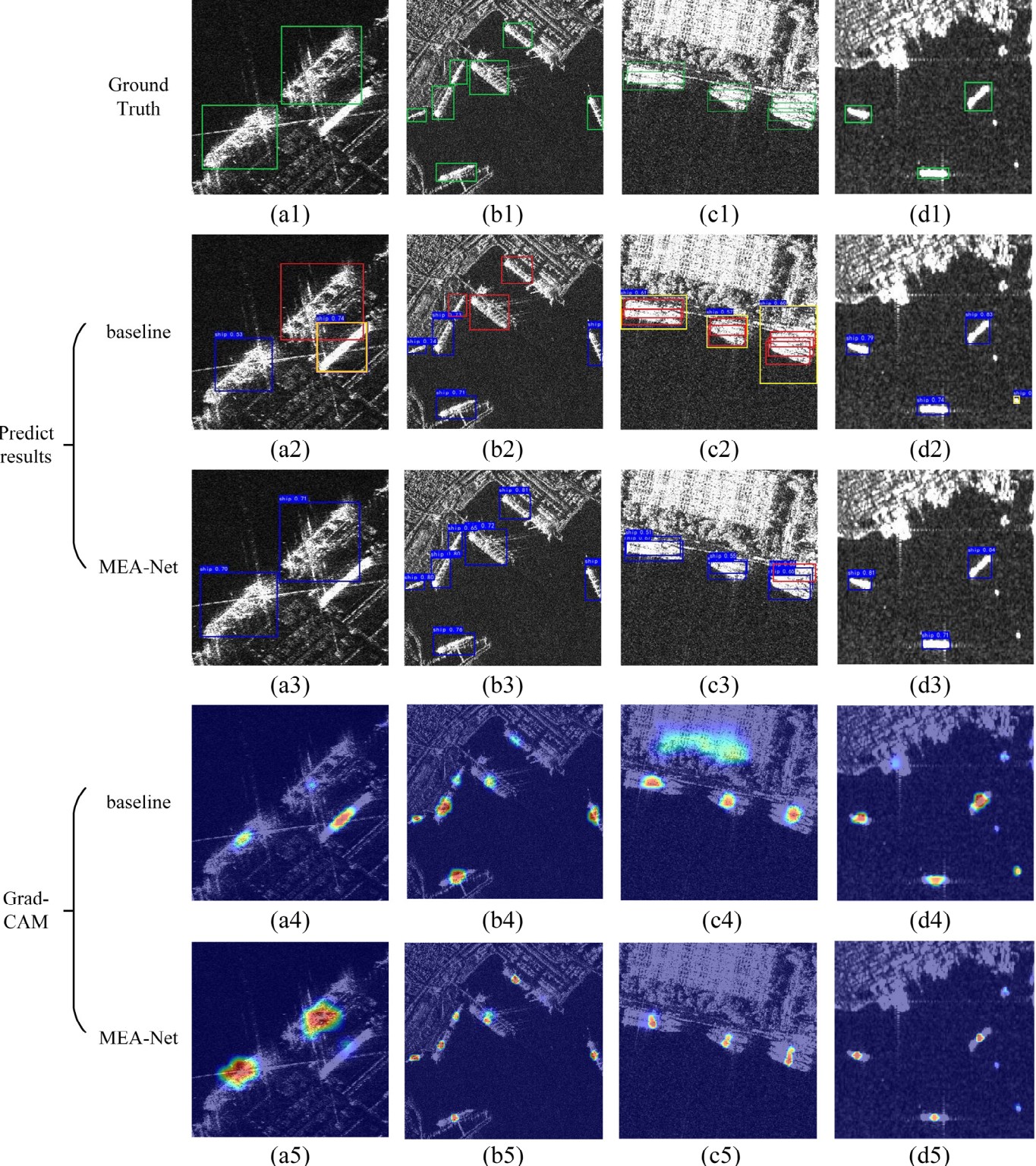

**Figure 8.** Comparison of visualization results between baseline and MEA-Net. (**a1**–**d1**): Ground truth. (**a2**–**d2**): Prediction results of baseline. (**a3**–**d3**): Prediction results of MEA-Net. (**a4**–**d4**): Activation heatmap of baseline. (**a5**–**d5**): Activation maps for MEA-Net. Green boxes represent truth boxes, blue boxes represent prediction boxes, red boxes represent false negatives, and yellow boxes represent false positives.

It can be seen from Figure 8(a2–d2,a3–d3) that, compared with the baseline model, the MEA-Net model improves the detection ability of large and medium inshore ship targets, dense ship targets, and offshore ship targets under the background noise. In Figure 8(a2), the baseline model identifies the land as a ship, and misses the detection of an inshore ship target. The reason for this problem can be explained by the corresponding activation heatmap Figure 8(a4). The baseline incorrectly points the interested region to land and weakens or loses the focus on inshore ship targets. In contrast, Figure 8(a5) demonstrates that the MEA-Net model pays more attention to the features of inshore ship targets and correctly focuses on the land region. For the detection in the dense ship scene, it can be seen from Figure 8(c2,c3) that the baseline model has a large number of false detections and missed detections. As explained by Figure 8(c4,c5), the baseline model cannot distinguish dense ship targets, while MEA-Net distributes the interested regions on each ship target. For the detection of offshore ship targets under the background noise, it can be seen from Figure 8(d2,d3) that the baseline model recognizes the noise points as a ship target, while the MEA-Net model recognizes all ship targets. This can be explained by Figure 8(d4,d5), the baseline model wrongly focuses on the noise points of the sea, while MEA-Net does not, and the focus region on the ship target is more concentrated than the baseline model.

### 4.8. The Comparative Experiments with Other Advanced Models

In order to verify the feasibility and generalization ability of the model proposed in this paper, we compare the performance of the MEA-Net model with the state-of-the-art models on the SSDD and HRSID datasets, respectively. Tables 6 and 7 are the experimental results on the SSDD and HRSID datasets, respectively. It can be seen that the amount of model parameters and FLOPs of MEA-Net is much lower than that of Faster R-CNN based on two-stage, CenterNet based on anchor-free, and several common lightweight models such as EfficientDet, YOLOv4-Tiny and YOLOv5-n, while reaching the highest *AP* value. The model parameters and FLOPs of MEA-Net are slightly higher than YOLOX-Nano by 0.06 M and 0.34 G, and the *FPS* is slightly lower by 0.21. However, the *AP* and *F*1 of MEA-Net improve, respectively, by 2.18% and 3.02% on the SSDD dataset, 3.64% and 3.70% on the HRSID dataset.

**Table 6.** Comparison of evaluation indicators of different models on the SSDD dataset.

| Model | *AP* (%) | *F*1 (%) | Params (M) | FLOPs (G) | *FPS* |
|---|---|---|---|---|---|
| Faster R-CNN | 73.09 | 65.87 | 136.69 | 258.58 | - |
| CenterNet | 90.56 | 85.06 | 32.67 | 79.12 | 1.34 |
| EfficientDet | 81.78 | 75.48 | 6.55 | 6.10 | 5.33 |
| YOLOv4-Tiny | 88.26 | 86.13 | 5.87 | 16.11 | 5.11 |
| YOLOv5-n | 91.91 | 89.06 | 1.90 | 4.50 | 5.83 |
| YOLOX-Nano | 90.65 | 89.90 | 0.90 | 2.46 | 6.52 |
| MEA-Net (ours) | 92.83 | 92.92 | 0.96 | 2.80 | 6.31 |

**Table 7.** Comparison of evaluation indicators of different models on the HRSID dataset.

| Model | *AP* (%) | *F*1 (%) | Params (M) | FLOPs (G) | *FPS* |
|---|---|---|---|---|---|
| Faster R-CNN | 77.65 | 77.10 | 136.69 | 258.58 | - |
| CenterNet | 84.30 | 72.71 | 32.67 | 79.12 | 1.34 |
| EfficientDet | 79.45 | 78.30 | 6.55 | 6.10 | 5.33 |
| YOLOv4-Tiny | 84.09 | 81.42 | 5.87 | 16.11 | 5.11 |
| YOLOv5-n | 87.24 | 83.73 | 1.90 | 4.50 | 5.83 |
| YOLOX-Nano | 85.42 | 82.73 | 0.90 | 2.46 | 6.52 |
| MEA-Net (ours) | 89.06 | 86.43 | 0.96 | 2.80 | 6.31 |

In order to further prove the superiority of MEA-Net, the detection effects of MEA-Net and other advanced target detection models are compared in the test images in the real scene. Here, green boxes represent truth boxes, blue boxes represent prediction boxes, red boxes represent false negatives, and yellow boxes represent false positives. Figure 9 is the comparison of the detection effects of different models for inshore medium ship targets on the SSDD dataset. It can be seen that MEA-Net detects all ship targets, while CenterNet, EfficientDet, YOLOv4-Tiny, YOLOv5-n, and YOLOX-Nano miss 2, 4, 6, 1 and 3 ship targets, respectively. Although Faster R-CNN also detects all of ship targets, there are two false detections. This can prove that MEA-Net can effectively suppress the influence of background to achieve accurate detection of inshore ship targets. Figure 10 is the comparison of the detection effects of different models for offshore small ship targets on the HRSID dataset. It can be seen that MEA-Net works best for offshore small ship detection compared to other models. Figure 11 is the comparison of the detection effects of different models for inshore large ship targets on the HRSID dataset. It can be seen that only MEA-Net detects all inshore large ship targets. The characteristics of the three large ship targets in Figure 11 are quite different, so many models cannot detect them all. However, MEA-Net relies on the excellent feature extraction ability of the EAM and the excellent multi-scale detection ability of the ARFB to precisely detect these ship targets.

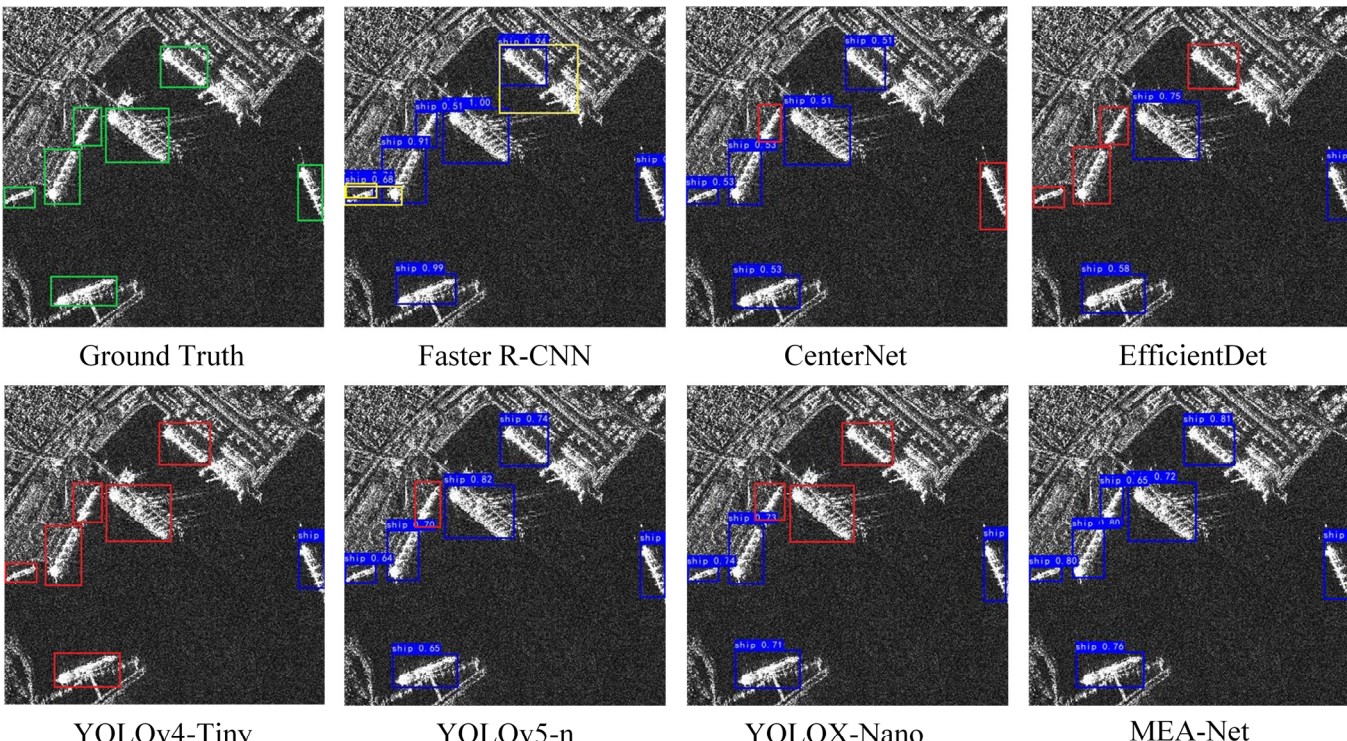

**Figure 9.** Comparison of the inshore medium ship detection effects of different models on the SSDD dataset. Green boxes represent truth boxes, blue boxes represent prediction boxes, red boxes represent false negatives, and yellow boxes represent false positives.

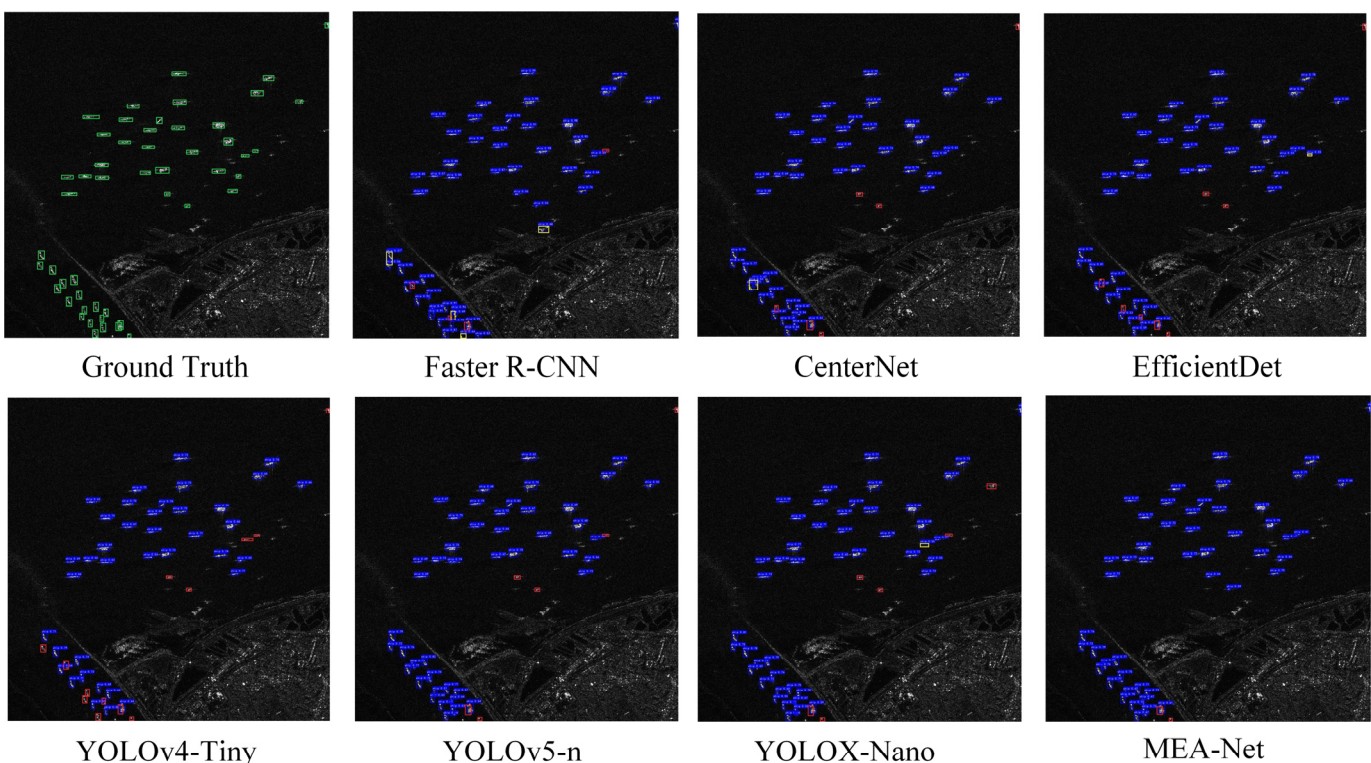

**Figure 10.** Comparison of the offshore small ship detection effects of different models on the HRSID dataset. Green boxes represent truth boxes, blue boxes represent prediction boxes, red boxes represent false negatives, and yellow boxes represent false positives.

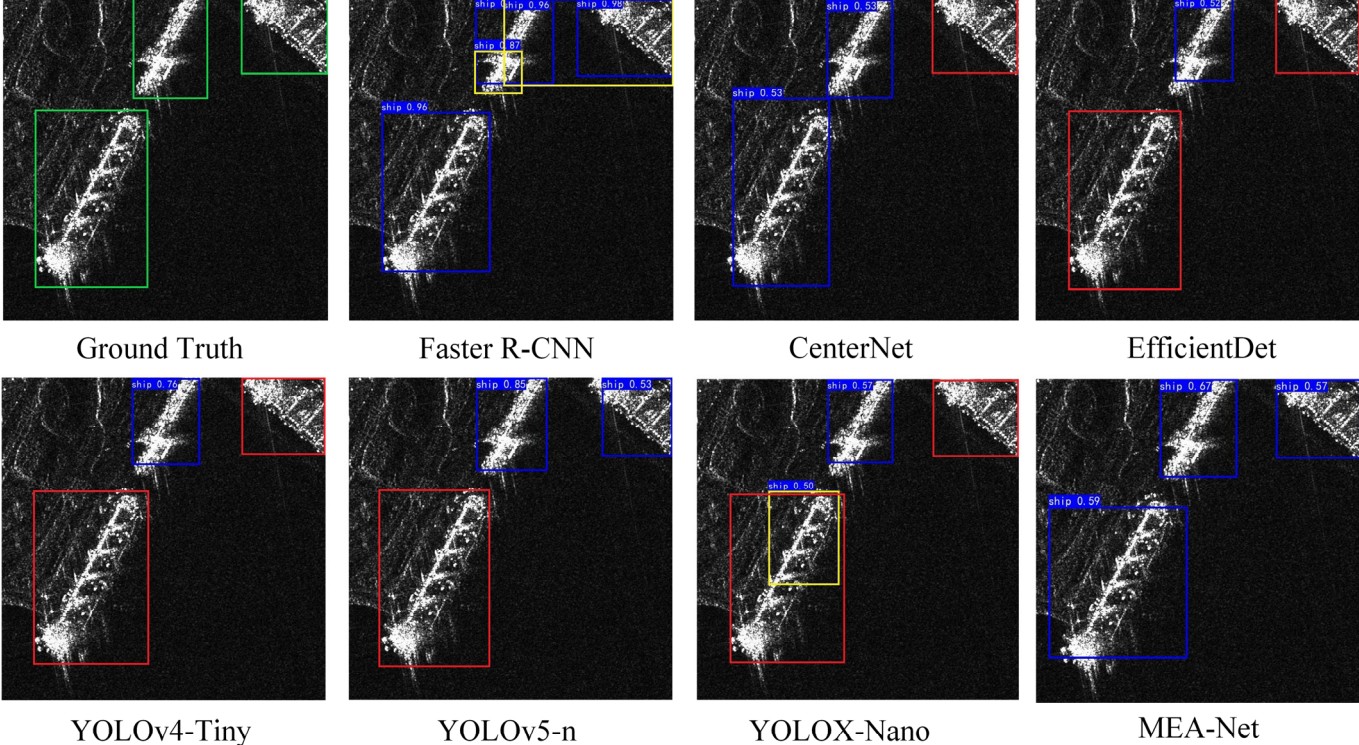

**Figure 11.** Comparison of the inshore large ship targets detection effects of different models on the HRSID dataset. Green boxes represent truth boxes, blue boxes represent prediction boxes, red boxes represent false negatives, and yellow boxes represent false positives.

## 5. Discussion

In the process of research, we find that the SAR ship datasets basically have the problem that the data volume of inshore and offshore ship targets do not match, and the data volume of ship targets of different sizes varies greatly. For the former, we innovatively propose a mask data balance augmentation (MDBA) method. By combining traditional image processing algorithms with XML label files in the dataset, the number of inshore ship targets can be effectively increased without affecting the number of offshore ship targets. For the latter, based on the traditional RFB-s multi-scale detection module, we propose an adaptive receptive field block (ARFB). Its adaptive convolution kernel size can improve the effect of multi-scale detection. At the same time, we find that there are very few ships target features in SAR images. In order to better extract ship features and improve detection accuracy without excessively increasing the complexity of the model, we propose an efficient attention mechanism (EAM). These proposed optimization modules have been verified to be effective and feasible, and good results have been obtained on both the SSDD dataset and the HRSID dataset.

## 6. Conclusions

In this paper, we propose a lightweight SAR ship detection model named MEA-Net for imbalanced datasets to solve the problem of large model structure, high computing resources, and poor detection results of inshore and multi-scale ship targets.

The following three optimizations are implemented. Firstly, we propose a data augmentation method named MDBA which solves the imbalance of inshore and offshore ship targets. Secondly, we propose an efficient attention mechanism module named EAM which can improve the feature extraction ability of the model for SAR ship targets. Thirdly, we propose an adaptive multi-scale detection module named ARFB which integrates local features and global features to achieve multi-scale detection. The above three optimizations are embedded into the baseline model at the same time. Compared with the baseline model, the *AP* and *F*1 of MEA-Net increase, respectively, 2.18% and 3.02% on the SSDD dataset, 3.64% and 3.70% on the HRSID dataset. The FLOPs and parameters of MEA-Net are only 2.80 G and 0.96 M, and the *FPS* reaches 6.31 when MEA-Net is deployed on the Jeston Nano edge computing device of the 2 GB version.

In future work, we continue our research to achieve more lightweight SAR ship detection, such as model pruning and knowledge distillation techniques. At the same time, the problem of imbalanced number of samples actually exists widely in different datasets, thereby the MEA-Net model proposed in this paper will be applied to other scenarios.

**Author Contributions:** Conceptualization, Y.G. and L.Z.; methodology, Y.G.; software, Y.G.; validation, Y.G.; resources, L.Z.; data curation, Y.G.; writing—original draft preparation, Y.G.; writing—review and editing, L.Z.; visualization, Y.G.; supervision, L.Z.; project administration, L.Z.; funding acquisition, L.Z. All authors have read and agreed to the published version of the manuscript.

**Funding:** This research was funded by the National Natural Science Foundation of China, No. 61901059 and No. 51978079.

**Data Availability Statement:** Not applicable.

**Conflicts of Interest:** The authors declare no conflict of interest.

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
