# Peer review of "MEA-Net: A Lightweight SAR Ship Detection Model for Imbalanced Datasets"

_remotesensing, doi:10.3390/rs14184438_

Round 1
Reviewer 1 Report
The method proposed in this paper is innovative and practically solves the existing problems. The three improvements proposed in this paper cover multiple stages of target detection, starting from data enhancement, feature extraction and feature enhancement, to improve the performance of the network more comprehensively. The paper is well-written with a good bond between theoretical developments and practical applications. The manuscript is a good one and I support it to be accepted for publication. However, for the methods and expressions in the article, there are some questions I hope the author can answer or modify. Once improved, I believe that the quality of the paper can be further enhanced.
1. In the MDBA method, when determine whether the SAR image contains land, why not directly assign 0 to the target area in the original image according to the label, but get a mask and then do the subtraction operation?
2. In the MDBA method, is it more effective to mask the offshore ships in the background containing land, or the enhancement of this part of the image later? Are there enhancements for all data in the baseline? If not, I recommended to add such an ablation study to further prove that the design of the MDBA method can improve the accuracy.
3. The MDBA method is mentioned in the abstract which can greatly improve the ability of the model to detect inshore ship targets. It is recommended to increase more experimental results for detecting inshore ships.
4. Please explain how are the two thresholds Tm and Tn determined in the MDBA method and is it a fixed value. In the second part, that is, the method of judging whether the ship target is inshore or offshore, suppose there are two scenes with land background, there are large inshore ships in one scene, and many tiny offshore ships in the other scene. The center points of the two kinds of ships are comparable in distance from the corresponding land boundary. At this time, if the threshold is fixed, will there be many wrong judgments? There are some such scenes in the HRSID dataset. Please explain if this problem is existing and how to solve it.
5. The contributions in the Introduction should be described in the order of the structure of the article.
6. In the abstract, it is stated that the problem to be solved is model lightweight. Please clearly state which one is more aimed at this problem in the following three innovations.
7. The EAM module is designed on the basis of CBAM and ECA, and it is said to be efficient in this paper. In subsequent experiments, please add these module to compare the parameters of the model to illustrate the role of EAM module in making model lightweight.
8. Please explain how are the parameters a and b determined to be 5000 and 400 when calculating k in the ARFB module.
Reviewer 2 Report
Dear authors
The manuscript is well prepared, the introduction is well written, and the conclusions summarize the findings of your research.
The methods are not described in details, but the flow sheets and your description explain in brief your algorithms. I suggest is future publication of your new research, to expand the section concerning the methods you utilized.
Some corrections to your paper:
- Line 288- 290: You write a kernel 7X7, I think you have to write the general case kXk and C2Dk in 288-289. Otherwise please explain why the kernel is of dimension 7.
Line 322-323. You write "we set a and b to 5000 and 400, respectively." that means a=5000, b=400
The values of a and b giving the results of k in Table 1 are : a=400, b=5000, that means the order is not correct. Please also explain in the text why you used these two values.
Reviewer 3 Report
This paper proposes a lightweight SAR ship detection model. But there are a few expressions listed following that need to be revised before publication.
1. Section 3.2, line 201. Please use capital letters at the beginning of a sentence.
2. The title of section 4.7, line 426. It is better to use the expression “visualization experiment”.
3. Some ship detection works are missing. For example, “Multi-Level Alignment Network for Cross-Domain Ship Detection, Remote Sensing, 2022”
Reviewer 4 Report
please see attached file.

Round 2
Reviewer 1 Report
In the response for comments 3, whether it is possible to add some comparisons of the results only for inshore ships to prove that MDBA improves the accuracy of inshore ship detection.
Reviewer 4 Report
I have no further comments.
